# CD44-Targeted Carriers: The Role of Molecular Weight of Hyaluronic Acid in the Uptake of Hyaluronic Acid-Based Nanoparticles

**DOI:** 10.3390/ph15010103

**Published:** 2022-01-17

**Authors:** Enrica Chiesa, Antonietta Greco, Federica Riva, Rossella Dorati, Bice Conti, Tiziana Modena, Ida Genta

**Affiliations:** 1Department of Surgery, Fondazione IRCCS Policlinico San Matteo, 27100 Pavia, Italy; 2Department of Drug Sciences, University of Pavia, 27100 Pavia, Italy; antonietta.greco@iusspavia.it (A.G.); rossella.dorati@unipv.it (R.D.); bice.conti@unipv.it (B.C.); tiziana.modena@unipv.it (T.M.); 3Department of Public Health, Experimental and Forensic Medicine, Histology and Embryology Unit, University of Pavia, 27100 Pavia, Italy; federica.riva01@unipv.it

**Keywords:** hyaluronic acid, CD44 targeting, microfluidic, nanoparticles, endocytic mechanism, transport inhibitors

## Abstract

Nanotechnology offers advanced biomedical tools for diagnosis and drug delivery, stressing the value of investigating the mechanisms by which nanocarriers interact with the biological environment. Herein, the cellular response to CD44-targeted nanoparticles (NPs) was investigated. CD44, the main hyaluronic acid (HA) receptor, is widely exploited as a target for therapeutic purposes. HA NPs were produced by microfluidic platform starting from HA with different molecular weights (Mw, 280, 540, 820 kDa) by polyelectrolyte complexation with chitosan (CS). Thanks to microfluidic technology, HA/CS NPs with the same physical features were produced, and only the effects of HA Mw on CD44-overexpressing cells (human mesenchymal stem cells, hMSCs) were studied. This work provides evidence of the HA/CS NPs biocompatibility regardless the HA Mw and reveals the effect of low Mw HA in improving the cell proliferation. Special attention was paid to the endocytic mechanisms used by HA/CS NPs to enter hMSCs. The results show the notable role of CD44 and the pronounced effect of HA Mw in the NPs’ internalization. HA/CS NPs uptake occurs via different endocytic pathways simultaneously, and most notably, NPs with 280 kDa HA were internalized by clathrin-mediated endocytosis. Instead, NPs with 820 kDa HA revealed a greater contribution of caveolae and cytoskeleton components.

## 1. Introduction

Over the last years, the development of nanotechnology to diagnose, screen, treat and prevent diseases has been significantly improved if it is considered that 43 nanomedicines have been approved by the European Medicines Agency (EMA), 71 by the US Food and Drug administration (FDA) and another 128 are currently in clinical trials [1,2].

Moreover, nanotechnology is expected to play an important role in providing new tools for various biomedical applications and create new solutions for the diagnosis and therapy of currently uncurable diseases [3,4]. However, the fulfilment of those expectations has been revealed more slowly than predicted because of several challenging issues, including regulatory and manufacturing aspects, as well as poor knowledge of critical quality attributes of nanomedicines essential for their activity and safety attributable to the poor scientific understanding of the nanomedicine interaction with the biological environment [5]. For the application of nanoparticles (NPs) as treatment of disease, the role of NPs targeting is crucial. Broadly, the accumulation of NPs in a diseased tissue can be achieved by “passive” or “active” targeting. The unique size of NPs is responsible for passive targeting. Owning to their nanometric dimensions NPs are able to pass through loose junctions and accumulate in a diseased tissue, but they suffer from several limitations such as lack of control in uptake and, indeed, being off-target drug delivery. Otherwise, active targeting exploits a targeting moiety (e.g., ligand, antibody, peptide) and its interaction with a receptor which is upregulated in a diseased tissue results in a less off-target delivery [6,7].

To use NPs to deliver drugs to a target tissue or organ, it is crucial to understand the endocytic mechanisms exploited by specific NPs to enter the cells, since these will likely affect the NPs’ final sub-cellular fate and localization [8,9,10]. Therefore, the therapeutic NPs’ efficacy could be increased if these nanosystems could be designed to reach a specific intracellular compartment (e.g., nuclei for gene therapy) [10]. Nowadays, several cellular uptake pathways have been described, and many efforts have been made to classify and characterize them [11,12]. Briefly, some features of the most common cellular uptake pathways are summarized hereafter.

Clathrin-mediated endocytosis involves a specific receptor that recognizes and internalizes the cargo into “coated pits” of 60–200 nm diameter and derived by the arrangement of a cytosolic coat protein named clathrin. These coated pits invaginate and pinch off to endocytic vesicles [13]. Caveolae-mediated endocytosis exploits the clustering of lipid raft components placed on the cellular membrane into flask-shaped infoldings named caveolae. It is induced by a specific ligand and considered the principal pathway of access for particles of about 200 nm [14]. Another uptake pathway classified as clathrin- and caveolae-independent endocytosis is macro-pinocytosis, which is liable to the internalization of large portion of cellular membrane with a large amount of fluid. It allows to internalize larger particles (diameter > 150 nm) by re-arranging the actin filament and creating a membrane protrusion [15]. As widely reported in literature, NPs’ physicochemical properties including, but not limited to, size, surface functionalization, shape and other factors such as concentration, time or cell types highly affect these uptake mechanisms [8,16,17]. Therefore, the study of the cellular uptake of specific NPs is complicated by the identification of several uptake routes and pathways. Moreover, the characteristics of the laboratory techniques, the inconsistency in some NPs stains used for tracking and the poor uniformity in NPs features allow the inconsistent findings regarding the specific endocytic route [18].

The present study aims to investigate the endocytic mechanisms exploited by Hyaluronic Acid/Chitosan NPs (HA/CS NPs) to enter human mesenchymal stem cells (hMSCs) for a rational development of a targeted drug delivery by understanding both surface binding and the internalization process. HA/CS NPs rely on their unique size and HA exposure on the surface to achieve a massive accumulation in diseased tissues through both passive and active targeting.

In HA-coated NPs, HA acts as targeting agent as it is the natural ligand of the CD44 receptor [19,20,21] and a therapeutically interesting target since it is upregulated in cancer-initiating or -metastasizing cells, with extensive involvement in the epithelial–mesenchymal transition, in carcer(-initiating) cell survival and drug resistance [21,22,23]. The use of HA to develop CD44-targeting carriers has also been stimulated by the numerous HA properties. HA is a biocompatible and biodegradable polymer, and it avoids the adsorption onto the NPs’ surface of most blood proteins, so it is useful for systemically administered NPs to prolong the NPs’ circulation time and prevent opsonization. The biological role of HA was widely studied during the last decade, and it resulted in being intensely influenced by the Mw [24,25].

In our previous work, HA/CS NPs were successfully produced by microfluidic technology, a relatively new technology for the reproducible and tunable synthesis of NPs for biomedical purposes [26,27]. A microfluidics system is a straightforward and effective tool for the synthesis of nanomaterials, ensuring a precise control over size, shape, and chemical composition through the manipulation of fluids in a microchannel. First, we developed a one-step synthesis of HA/CS NPs based on electrostatic interactions between HA and CS. Then, the role of HA Mw in the NPs’ assembly into the microfluidic channel has been specifically faced. Finally, a relationship between polymer Mw and solution viscosity has been proposed to maintain specific quality attributes (size, PDI, surface charge) to NPs [26,27]. The feasibility and reliability of the HA/CS NPs as a drug delivery system have been proven by our research group for different payloads, both in lipophilic synthetic drugs (Everolimus) and macromolecules (Myoglobin). However, many studies regarding HA/CS NPs as carrier for drugs were reported in the literature, especially for gene delivery [28,29,30,31,32].

This work could provide new insights on the nature of the uptake mechanism of HA/CS NPs, prepared by a microfluidics technique. It is worth mentioning that the internalization of HA/CS NPs seems related to the different HA presentations, affecting the affinity and speed of uptake, as well as the cytotoxicity, with strongly cell-dependent effects [28,33]. More recently, HA-mediated targeting has been interpreted as a more complex phenomenon of endocytic recognition rather than a cell surface event of selective binding [28].

More in detail, in this study, the impact of HA average molecular weight (Mw) was investigated while the influence of the NPs’ size was avoided; three HA polymers with different Mw were tested, namely, 280, 540 and 820 kDa, and the size of the NPs was kept constant at around 200 nm. hMSCs from human bone marrow were used because they are well-known CD44-positive cells [34] largely localized in solid and desmoplastic tumors.

The cellular uptake HA/CS NPs was systematically studied to assess: (i) the energy dependance of the NPs uptake by incubating the hMSCs at 4 °C; (ii) the role of HA–CD44 interactions by treating hMSCs with mouse antiCD44 glycoprotein primary antibody before the exposure to NPs to prevent HA binding to CD44; (iii) the mechanism of HA/CS NPs internalization (clathrin vs. caveolin) by using specific pharmacological inhibitors (Chlorpromazine chloride and Genistein); and (iv) the effect of cytoskeleton components such as microtubules and F actin on the HA/CS NPs uptake by the treatment with Nocodazole and Cytochalasin D.

## 2. Results

### 2.1. HA/CS NPs Preparation and Characterization

CD44 is an endocytic HA receptor that is overexpressed in several types of carcinomas and involved in many physiological and pathological pathways [35]. For this reason, the use of HA to achieve CD44 targeting carriers has been encouraged [29,36,37,38,39]. In this paper, we aim to highlight if a relationship exists between HA Mw and the CD44-mediated NPs internalization and extent of binding. As widely known in literature, the nanocarriers’ physicochemical properties mainly influence their uptake into the target cells [16]. To avoid any interference of NPs features and point out only the role of HA Mw on the NPs’ uptake mechanism, HA/CS NPs made of different HAs (280, 540, 820 kDa Mw) but with similar sizes, size distributions and surface charges were produced. HA/CS NPs were prepared by one-step polyelectrolyte complexation through a microfluidic device equipped with an SHM micromixer. Please refer to our previous paper for a detailed study of microfluidic production conditions [27]. The HA/CS NPs made of HA with an Mw of 280 kDa presented a mean size of 211 ± 19.3 nm with a PDI of 0.30 ± 0.02 and a negatively charged surface (−16.6 ± 2.21 mV). By using HA with an Mw of 540 kDa, the NPs had a mean diameter of 177 ± 20.2 nm (PDI = 0.30 ± 0.03), and they were negatively surface charged (−19.7 ± 5.35 mV). Lastly, starting from the HA with the highest Mw (820 kDa), the HA/CS NPs showed a mean size of 197 ± 27.1, a PDI of 0.29 ± 0.01 and a surface charge of −19.5 ± 2.16 mV.

A Tukey’s multiple comparisons statistical analysis revealed no size differences (*p* value > 0.05) among the formulations obtained from HAs with different Mw. Moreover, the consistent negative charge relied on the HA deposition onto the NPs’ surface that is crucial for the NPs’ interaction with CD44 to trigger the NPs’ uptake.

A TEM analysis (Figure 1a) performed on the same NPs confirmed the DLS-detected diameters of around 200 nm, further showing spherically shaped NPs for all the HAs tested.

NPs trafficking through the cell membrane was followed by fluorescent HA/CS-RhB NPs. With this purpose, CS was grafted to RhB (CS-RhB) by using the RhB carboxylic group and the amine groups on CS chain with a reaction yield of 45%.

Figure 1b shows the comparison between the UV-vis spectra of the CS-RhB conjugate solution (5 µg/mL, dashed line) and that of RhB solution (5 µg/mL, full line), proving the CS-RhB conjugation. The RhB labeling efficiency was evaluated by measuring the absorbance at 554 nm of the CS-RhB conjugate against RhB standard solutions ranging from 0.25 to 4 µg/mL, and it was of 0.21 ± 0.03 µg of RhB/µg of CS.

The HA/CS-RhB NPs showed mean diameters of 249 ± 13.4 nm, 183 ± 43.3 nm and 235 ± 26.5 by using HA with Mw of 280, 540 and 820 kDa, respectively. No differences were highlighted by using Tukey’s multiple comparison test. The PDI value is around 0.3 for all the formulations, and negative surface charges of about −16.6 ± 2.21, −19.7 ± 2.28 and −19.5 ± 2.16 mV for HAs with 280, 540 and 820 kDa Mw confirmed the HA localization on the NPs outer surface.

### 2.2. Cytotoxicity

HA and CS are commonly selected and used to produce drug delivery systems because of their biocompatibility and biodegradability [25,40]. MTT assay results prove that placebo HA/CS NPs were not cytotoxic up to 1000 µg/mL concentration, and the well-established biocompatibility of HA/CS NPs was confirmed since the cell viability percentage was never less than 80% (Figure 2a). The quantitative results were supported by morphological analysis. As shown in Figure 2b, placebo HA/CS NPs did not trigger any toxic effects because no cell morphology alterations were observed at the end of the incubation.

### 2.3. Proliferation Study

HA plays a key role in biology as a structural and signaling molecule. Its activity is strongly influenced by the Mw and the physiological/pathological environment. Despite the enormous efforts of many researchers to figure out the biological roles of HA, only its general biological behavior has been identified so far. Furthermore, opposite effects have been reported for high and low HA Mw. According to literature, HA with a high Mw has an anti-inflammatory and immunosuppressive action and promotes wound healing; on the other hand, HA with a low Mw (including HA fragments) increases the production of proinflammatory factors and supports ECM remodeling and tumor progression [41,42]. The impact of HA Mw on hMSCs proliferation was evaluated by monitoring the DNA synthesis through the 5-bromo-20-deoxyuridine (BrdU) nuclei incorporation method.

As shown in Figure 3a, at 4 h of incubation, no significant effect on the cell proliferation was highlighted for all the HA Mws tested. With respect to the untreated cells (CTRL), a slight increment in the proliferative cell number was observed after the treatment with 180 μg/mL of HA/CS NPs with HA of 280 kDa. Prolonging the incubation time at 24 h (Figure 3b), a notable increase in the proliferative cells was revealed by using the lowest HA Mw at both NPs concentrations (10 and 180 μg/mL). No differences were pointed out by using HA/CS NPs made of HA with Mws of 540 and 820 kDa. It is worth mentioning that the proliferation study results were mainly influenced by the cell line used: similar conclusions were achieved by Zhao et al. (2015) by using rabbit bone-marrow-derived stem cells and testing different HA solutions [43].

### 2.4. Uptake Study

To evaluate the interplay between the HA Mw and the binding and uptake of the HA/CS-RhB NPs and whether NPs uptake follows an active or passive process, the hMSCs were incubated at 37 °C or 4 °C for 90 min with 100 μg/mL of fluorescent HA/CS-RhB NPs. This time point was chosen from our previous results and based on evidence found in literature, where it is shown that the HA-based NPs’ uptake reached a plateau already at 2 h of incubation for several cell models [26,30].

We evaluated the fluorescence intensity of NPs internalized by cells at 90 min incubation time, and the results are shown in Figure 4. As highlighted in confocal images, all HA/CS NPs were successfully internalized by hMSCs: They are widely distributed in cytoplasm cell compartment (Figure 4a) as well as enclosed in cellular membrane vesicles located near the cell surface (Figure 4b). As far as untreated control cells, as expected, no red fluorescence inside the cells was detected. The elaboration of the confocal images along the z-axis allows us to obtain the 3D project of the hMSCs incubated with NPs and, as it can be seen in Figure 4c, the HA/CS-RhB NPs were near the perinuclear region surrounded by the cellular membrane.

After that, confocal images were processed by ImageJ software to quantify the fluorescence level inside the cells, and quantitative results are reported in Figure 5.

Tukey’s multiple comparison test did not reveal any differences between HAs with Mws of 280 and 540 kDa; instead, the cellular uptake of HA/CS-RhB NPs made of HA with Mw of 820 kDa was notably reduced (*p* value < 0.001).

Exposure to NPs at 4 °C resulted in a very strong inhibition of the endocytosis, as visualized by the lack of red fluorescence in Figure 4a. The reduction of intracellular fluorescence level was of 66.6, 77.9 and 94.9% for NPs made of HA with Mws of 280, 540 and 820 kDa, respectively (Figure 5), indicating that NPs uptake is extremely energy-dependent. Although the cell viability and the cell behavior may be compromised by lowering the temperature of the cell culture, we demonstrated by morphological examination that hMSCs growth was not altered at the end of the incubation, and thus we proved that the decreased uptake is a consequence of the active transport inhibition. In literature, a trend has been recognized in the inhibition due to different NPs sizes where a stronger reduction in the energy dependence condition was shown for smaller NPs; however, here, the result is attributable only to the HA Mw.

### 2.5. Competitive Binding Experiment

To evaluate whether the CD44–HA interaction plays a role in the NP uptake, competitive binding experiments were performed by pre-treating hMSCs with mouse anti-CD44 glycoprotein primary antibody (antiCD44) and enabling them to prevent HA binding to CD44 and thereby NPs internalization. As can be noted from Figure 6, the incubation with antiCD44 notably reduced the NPs cell internalization; a cellular uptake inhibition of 80.2%, 91.6% and 82.3% was reached by using HA/CS-RhB NPs prepared with HA of 280 kDa, 540 kDa and 820 kDa, respectively. After the pre-treatment with antiCD44, the fluorescence level inside the cells was almost negligible, and no differences were highlighted among the different HA Mws. These results prove that the internalization of HA/CS NPs into hMSCs primarily occurs by a specific interaction with CD44 regardless of the HA Mw used.

### 2.6. Transport Inhibition of HA/CS NPs Uptake

A deep understanding of the mechanism by which HA/CS NPs are internalized by cells is of great importance to optimize the efficacy of NPs delivery to cells. Gaining insight into the role of various endocytic pathways to the NPs’ cellular uptake can be useful to achieve this purpose because it will allow us to correlate the carrier’s physicochemical properties with cellular uptake, intracellular processing and fate. Here, we have used specific chemical inhibitors to explore their capability to selectively prevent some of the major endocytic pathways. Genistein and Chlorpromazine hydrochloride selectively block the caveole- and clathrin-mediated endocytosis, respectively. Moreover, the cytoskeleton has a significant role, being involved in endocytosis as well as the trafficking of endocytosis vesicles (mainly macro- and pinocytosis); therefore, the role of F-actin and microtubules in the uptake of HA/CS NPs was investigated by using Cytochalasin D and Nocodazole. Cytochalasin D is an inhibitor of F-actin polymerization, while Nocodazole disrupts the microtubules.

The results are shown in Figure 7a. The uptake of the HA/CS NPs made of HA with a Mw of 280 kDa was dramatically reduced by pre-treating cells with Chlorpromazine hydrochloride if compared with untreated cells cultured at 37 °C (*p* value < 0.0001, Figure 4a). Instead, no statical differences were revealed when cells were treated with other inhibitors.

By preventing the clathrin disassembly and receptor recycling to the cell membrane, Chlorpromazine triggered about 75.6% reduction in HA/CS NPs uptake when 280 kDa HA was used to prepare the NPs. For this reason, we can assume that the HA/CS NPs made of 280 kDa HA were mainly internalized by the cell through a clathrin-mediated endocytosis. No dependence to caveolae and cytoskeleton activity was highlighted.

Increasing the HA Mw to 540 kDa, the HA/CS NPs’ uptake resulted in being more unspecific, and all the endocytic pathways seemed to be involved. Genistein, which is a specific inhibitor of tyrosine kinases involved in caveolae-mediated endocytosis, reduced the uptake of HA/CS NPs by 89.6%. The level of inhibition was similar by using Chlorpromazine that caused a decrease in the uptake of HA/CS NPs by 84.4%. Therefore, using 540 kDa, HA/CS NPs exploit both caveolae- and clathrin-mediated mechanisms to be internalized by hMSCs. Finally, as can be seen from Figure 7b, the cytoskeleton was involved in the uptake of HA/CS NPs that was reduced by 86.8% and 69.7% by using Cytochalasin D and Nocodazole, respectively, indicating that macro- and pino-cytosis should also not be neglected. Moreover, this indicates a stronger contribution of actin filaments for the internalization of the NPs with respect to microtubules for 540 kDa HA.

Interestingly, by using HA with a Mw of 820 kDa, the internalization of the NPs was not influenced by Chlorpromazine, and no statistical differences were revealed if compared to untreated cells (37 °C, Figure 7c). On the other hand, the treatment with Genistein reduced the uptake of NPs by 76.9%, and indeed, the internalization of HA/CS NPs occurred preferentially through caveolae-mediated endocytosis. Moreover, a greater cytoskeleton contribution was observed because NP uptake inhibitions of 96.6% and 90.3% were obtained by using Cytochalasin D and Nocodazole, respectively. In this case, F-actin and microtubules had comparable roles.

These data were confirmed by confocal images in Figure 8, where nuclei were shown in blue, HA/CS NPs in red and CD44 were expressed on the hMSCs’ membrane in green.

### 2.7. Morphological Examination

One important requirement when endocytosis inhibitors are used is that they should not alter the normal cells functions and not directly involve actin, thereby confounding the data and leading to multiple simultaneous effects. Therefore, evaluating the inhibitors’ cellular toxicity is crucial. At the end of the treatment with the different pharmacological inhibitors, a morphological examination of hMSC did not reveal any significant cell alteration. Moreover, the cytoskeleton organization was visualized by highlighting actin filaments with Phalloidin-FITC (Figure 9, nuclei in blue, actin in green): It can be observed that the actin filaments maintained their general structure when treated with inhibitors, excluding Cytochalasin D, which was used with the specific aim to disrupt actine filaments during the NP uptake process.

## 3. Discussion

Since its first isolation, HA has been studied across a variety of research areas because of its unique physicochemical properties. HA application for the design of nanosized drug delivery systems is one of the most investigated since HA has outstanding characteristics as a nanocarrier, and it is an active targeting moiety of NPs. HA is a ligand of the CD44 receptor, which is overexpressed on the cell membranes of several types of tumors. This work aims to dissect the mechanistic role of CD44 and the endocytic pathways used by HA-based NPs to enter hMSCs, massively found in solid and desmoplastic tumors, where CD44 is upregulated.

Several methods can be employed to prepare HA-based nanoparticles, the most popular being the ionotropic gelation [29,44,45], where the driving force is the electrostatic attraction between multiple positively charged cations and a deprotonated carboxylic group of HA. CS is usually used as a crosslinker, and its main advantages are its low toxicity and its biodegradability. Ionotropic gelation is a gentle manufacturing process, performed in aqueous and quite similar physiological environments and without chemical reactions [28].

Here, HA/CS NPs were successfully prepared by one-step polyelectrolyte complexation through a microfluidic device with an SHM micromixer by mixing CS and HA/TPP solutions. Likewise, in bulk ionotropic gelation techniques, ancillary amounts of TPP allow obtaining a suitable nanostructure organization and compactness, thereby ensuring the proper NPs formation via CS/TPP ionic gelation [46].

Aiming to evaluate only the effect of HA Mw on the NPs’ uptake, NPs with comparable physical features were produced by HA having different Mw, namely, 280 kDa, 540 kDa and 820 kDa. By exploiting the methodological approach proposed in our previous work [27], the recovered HA/CS NPs showed particles size of around 200 nm, a PDI < 0.3 and negative surface charges. The size and size uniformity agree with the most common nanomedicine applications, meanwhile, the negative surface charge indicates the presence of HA on the NPs surface that is crucial for the CD44 targeting.

Microfluidic technology demonstrated to be a versatile tool to produce high-quality NPs under a less hardworking process regardless the characteristics of the starting materials.

HA/CS NPs are confirmed to be biocompatible and safe nanocarriers for drug delivery because no toxic effect was revealed up to a concentration of 1000 µg/mL regardless the HA Mw used. Since cell viability percentage was always greater than 80%, IC50 values were higher than 1000 μg/mL for all the HA tested. Moreover, HA Mw influence on the hMSC proliferation was highlighted by assessing the BrdU incorporation. The lowest HA Mw caused a slight increase in hMSC proliferation if used at the highest concentration (180 µg/mL) after 4 h of incubation, and this enhancement was notably higher if the incubation was prolonged at 24 h for both the concentrations tested (10 and 180 µg/mL). No effect on the hMSC proliferation was observed for HAs with Mws of 540 and 820 kDa. These results confirm the complex biological role of HA, which is mainly influenced by the Mw. However, the investigation of the relationship that interplays between the HA’s Mw and its biological function is difficult because it is strictly correlated to the panel of cell lines used. Our results were supported by Zhao et al. (2015) who tested an aqueous solution of HA with different Mws (low, medium and high) on rabbit bone-marrow-derived stem cells. The authors reported that HA with low Mw triggered an increase in cell proliferation; meanwhile, by using medium and high HA Mws, the proliferation remained unchanged, or slightly reduced [43]. Of course, evidence on hMSC proliferation may be more noticeable after a long-term treatment (7–14 days), however, for a proper evaluation of the HA/CS NPs’ long-term effect, both multiple doses of the NPs and the proper frequency of administration should be considered based on of the definite application. Therefore, future studies will address these specific issues.

To follow NPs trafficking into hMSCs, CS was grafted with RhB, and fluorescent NPs were obtained without significant differences if compared to blank HA/CS NPs. To investigate by which endocytic mechanisms HA/CS NPs are internalized by hMSC, the uptake of NPs was systematically studied by confocal microscopy. Confocal images were elaborated by ImageJ to obtain quantitative data. NP uptake was evaluated at short exposure time since it was previously reported that HA/CS NP uptake reached a plateau at 2 h of incubation [30]. Furthermore, blocking one uptake pathway can activate an alternative endocytic mechanism, which confounds the data [8].

As expected by incubating hMCS and HA/CS NPs at 37 °C for 90 min, the NPs were massively internalized by the cells, they were mainly located in the cells’ cytosol, and endocytic vesicles were highlighted near the plasma membrane. At 90 min of incubation, the uptake of HA/CS NPs made of 820 kDa HA is slower if compared with that of NPs composed of HA of 540 kDa and 280 kDa. Comparable results were recently reported by Della Sala et al. (2022) [47] who studied the uptake kinetic of HA-coated PLGA NPs by using different HAs (200 kDa, 800 kDa and 1450 kDa), and they pointed out that at 1 h of incubation, the NPs uptake was lower if HA with an Mw of 200 kDa was used, while after 6 h of exposure, the uptake of NPs made of HA Mws of 200 and 800 kDa were similar. Several mechanisms can be liable to the different HA–CD44 binding. First of all, HA’s affinity to the receptor can change due to CD44 glycosylation [21,48]. Then, a single HA chain contains multiple CD44 binding sites, thereby it can interact with several CD44 receptors at the same time [48]. Finally, HA with a high Mw can assume several arrangements on the NP surface because of its chain flexibility.

Lowering the incubation temperature to 4 °C caused a very strong inhibition of the NPs uptake for all the NPs tested. Moreover, the pivotal role of HA in the internalization of HA/CS NPs was demonstrated by a competitive binding experiment when hMSCs were treated with mouse antiCD44 glycoprotein primary antibody (antiCD44). Regardless of the HA used, the NPs’ uptake was deeply reduced, reaching an inhibition percentage of around 80%. Different endocytic pathways can occur, and they were studied by specific pharmacological inhibitors. It is important for these experiments that transport inhibitors ensure the endocytosis reduction without causing too-severe cellular side effects.

The results disclosed that the HA Mw impacts the endocytic mechanism used by the NPs to enter hMSCs. HA/CS NPs with HA of 280 kDa are mainly internalized by the cell through a clathrin-mediated endocytosis. No dependence to caveolae and cytoskeleton activity was highlighted. Increasing the HA Mw to 540 kDa, the HA/CS NPs’ uptake resulted in being more unspecific, and all the endocytic pathways seem to be involved. Finally, HA/CS NPs with HA of 820 kDa are preferentially internalized trough caveolae-mediated endocytosis with a greater cytoskeleton involvement.

The results presented here highlight the complexity of the mechanisms that cells use for NP endocytosis and suggest that different pathways can simultaneously occur to internalize the same NPs and that the raw materials’ features influence the mechanism used. These results indicated the importance of careful analysis and individual interpretation for each NP and each cell line. Because of this work, future studies will be focused on spreading the cell lines panel to investigate the avidity of HA–CD44 interaction because HA can discriminate between different CD44 densities, leading to possible off-target effects. Moreover, CD44 identity will be taken in consideration since CD44 is present in different isoforms with several post-modifications that can affect the interaction with HA. More in detail, cell line panels will be extended to include human cancer cells as fundamental models to assess the therapeutic efficacy of an anticancer drug. Special attention will be paid to pancreatic adenocarcinoma (PDAC), where CD44 is well-known for contributing to pancreatic cancer cell plasticity, invasiveness and response to therapy. Future in vivo studies will be planned to test HA/CS NPs on an orthotopic PDAC mouse model.

## 4. Materials and Methods

Hyaluronic acids (HA, Mw 280, 540 and 820 kDa) were provided by Faravelli SpA (Milano, Italy). Chitosan Chloride salt pharmaceutical grade (CS, Chitoceuticals, viscosity 19 mPa (1% in water), Mw 110 kDa, chloride content 13%, deacetylation degree 82.2%) was purchased from Heppe Medical Chitosan GmbH (Halle, Germany). Sodium tripolyphosphate (TPP) (technical grade 85%, Mw = 367.86 g/mol), Chlorpromazine Hydrochloride, Cytochalasin D, Genistein, Nocodazole, Dulbecco’s Modified Eagle’s Medium High glucose (D-MEM), Dulbecco’s Phosphate Buffered Saline (PBS 10x, sterile), Trypsin-EDTA, Dimethyl sulfoxide (DMSO), Anti-mouse IgG secondary antibody Fluorescein IsoThioCyanate (FITC) conjugated, Hoechst 33258 solution and Penicillin-Streptomycin were all from Sigma-Aldrich (St. Louis, MO, USA). Phagocyticglycoprotein-1, monoclonal mouse primary antibody (antiCD44) anti-mouse IgG secondary antibody FITC conjugated were purchased from Biogenex (San Ramon, CA, USA). Monoclonal Anti-BrdU antibody produced in mouse was obtained from Amersham-GE-HealthCare UK Limited (Buckinghamshire, UK). Fetal bovine serum was obtained by EuroClone Spa. If not specify, distilled filtered water was used (Millipore Corporation, Billerica, MA, USA), and solvents of analytical grade were used.

### 4.1. Cell Lines

hMSCs from human bone marrow were gifted by the Department of Public Health, Experimental Medicine and Forensic, Histology and Embryology Unit, University of Pavia.

### 4.2. HA/CS NPs Preparation

NPs were synthetized through the microfluidic platform NanoAssemblr^TM^ Benchtop (Precision NanoSystems Inc., Vancouver, BC, Canada). The microfluidic mixer has a characteristic Y-shape with staggered herringbone structures (SHM) [49]. HA/TPP and CS aqueous solutions were pumped into separate inlets of the microfluidic device. As demonstrated by our previous work [27], depending on the HA Mw, the HA and CS concentration were properly changed as reported in Table 1. The TPP concentration was kept constant at 2 μg/mL, and it was used as ancillary crosslinker to trigger the CS nucleation. Microfluidic process parameters were set as follows: flow rate ratio (FRR) 1:1, total flow rate (TFR) of 12 mL/min for all the formulations produced. A sample of 3 mL was collected from the output by setting a start and end waste of 0.350 and 0.050 mL, respectively.

A/CS NPs were purified by centrifugation (16,400 rpm, 4 °C, 30 min) (Eppendorf Centrifuge 5417 R, Eppendorf s.r.l., Milan, Italy). NPs were then resuspended and diluted at the proper concentration into DMEM used for cell culture at room temperature.

### 4.3. Rhodamine B Labelled NPs Preparation

An amidation reaction was exploited to graft Rhodamine B (RhB) to CS by using 1-Ethyl-3-[3-(dimethylamino) propyl] carbodiimide hydrochloride (EDC) and N-hydroxysuccinimide (NHS) as catalyzers, following the protocol previously described in [49]. Product purification was performed by dialysis (MWCO: 12–14,000 Da, SPECTRA/POR^®^) against water for 3 days, avoiding light exposure. Finally, the purified product was freeze-dried, and the reaction yield was gravimetrically determined. Labeling efficiency was assessed by UV-vis spectroscopy by measuring the absorbance of CS-RhB conjugate solution against RhB standard solutions ranged from 0.25 to 4 µg/mL (R^2^ = 0.99). The wavelength was set at 554 nm.

Fluorescent HA/CS-RhB NPs were produced through the microfluidic method described previously starting from a blend of CS (90% *w*/*w*) and CS-RhB (10% *w*/*w*).

### 4.4. HA/CS Nanoparticles Characterization

Physical properties of HA/CS NPs (mean diameter, PDI and surface charge) were assessed by dynamic light scattering (NICOMP 380 ZLS, Particles Sizing System, Santa Barbara, CA, USA). All the analyses were performed in triplicate for each formulation, and results were shown as mean ± SD. Morphological examination of NPs was performed by transmission electron microscopy (TEM) (JEOL JEM-1200EXIII with TEM CCD camera Mega View III, Tokyo, Japan) by using a negative staining (1% *w*/*v* uranyl acetate); NP sizes were determined by processing the TEM images by ImageJ software [50].

### 4.5. Cytotoxicity

NP cytotoxicity was investigated by MTT (3-[4,5-dimethylthiazol-2-yl]-2,5-diphenyltetrazolium bromide) assay. Briefly, hMSCs were seeded in 96-well plates at the density of 10,000 cells/well. Then, hMSCs were exposed to increasing concentrations of HA/CS NPs, ranging from 12.5 to 1000 µg/mL for each HA Mw, and incubated for 24 h. At the end of incubation, cells were washed by sterile PBS prior to the incubation with MTT working solution (5 mg/mL) for 2.5 h. Afterward, cell membrane was solubilized by DMSO to allow the dissolution of formazan crystals. Cell viability values resulted from formazan crystals absorbance detection at 570/690 nm through a plate reader device (SpectraMax M2e, Molecular Device LLC, San Jose, CA, USA); results were expressed as cell viability percentage (mean ± SD, n = 3). Untreated cells were used as control (CTRL).

### 4.6. Assessment of DNA Synthesis and Proliferative Activity by 5-Bromo-20-Deoxyuridine Incorporation

Cells proliferation was evaluated by monitoring the DNA synthesis through the 5-bromo-20-deoxyuridine (BrdU) nuclei incorporation method. Briefly, 20,000 cells were seeded on rounded glass bottom dishes and incubated for 24 h. Then, they were treated with 200 µL of HA/CS NPs resuspended DMEM with 10% *v*/*v* FBS at the concentration of 10 and 180 µg/mL and incubated for 90 min. Cells without NP treatment were used as negative control. At expired time, cells were incubated with 500 µL of 30 mM BrdU solution for 45 min; cells were then fixed in paraformaldehyde 4% *w*/*v* aqueous solution. The incorporated BrdU was labeled as described in [26].

Hoechst33258 solution (0.5 µg/mL) was used to stain hMSCs’ nuclei. Samples were visualized by fluorescence microscope (Zeiss Axiophot, Carl Zeiss, Oberkochen, Germany; blue filter: λ ex. = 346 nm and λ em = 460 nm; green filter: λ = 494 nm and λ em = 518 nm). The proliferation activity percentage values of hMSCs treated with HA/CS NPs were calculated from the number of proliferating cells (BrdU positive nuclei) against the total number of viable cells; results were compared with the negative control.

### 4.7. Cellular Uptake of HA/CS NPs

HA/CS-RhB NPs internalization in hMSCs was tracked by confocal microscopy (Leica TCS SP8, AOBS, Milan, Italy). hMSCs (20,000 cells /well) were seeded on glass slides and cultured in DMEM with 10% *v*/*v* FBS and 1% *v*/*v* antibiotics at 37 °C, 5% CO_2_. Cells were exposed to HA-RhB NPs (100 µg/mL) for 90 min, as reported in our previous work [26]. Afterwards all culture media were removed, and hMSCs were washed with sterile PBS buffer.

Energy dependence experiments were performed by lowering the cell culture temperature to 4 °C for 30 min prior to adding the NPs. After this preincubation, the treatments with HA/CS-RhB NPs (100 µg/mL) were carried out for 90 min at 4 °C. At the scheduled time points, the medium was discarded, and cells were washed with PBS to successfully remove NPs from the outer cell membrane.

Finally, hMSCs were fixed with 4% (wt) paraformaldehyde. Immunocytochemistry assay (ICC) was used to figure out the CD44 on hMSCs, following the protocol stated in [51]. Cell nuclei were dyed by Hoechst33258. Cells were examined by confocal microscopy (obj mag 40×).

The fluorescence level due to the NPs inside the cells was assessed by elaborating the confocal images by ImageJ software Version 1.52a, thus obtaining a quantitative outcome of the fluorescent NPs uptake into hMSCs. On each glass slide, 10 confocal images were taken by isolating the red fluorescence (λ_exc_ = 559 nm; λ_em_ = 580 nm). The cell of interest was selected by using the drawing tools (free form) of the software, and ImageJ provided data about Area, Mean fluorescence and Integrated density of the selection. Then, a region next to the cell that had no fluorescence was selected; this was the background. This procedure was repeated for at least 6 cells in the field of view. Minimum 60 different cells per sample were analyzed.

By Equation (1), results were stated as Corrected Total Cell Fluorescence (CTCF), calculated as:CTCF = Integrated density − (Area of selected cell × Mean fluorescence of background readings)(1)

Data were graphically represented as CTCF percentage compared to the control, namely, hMSCs treated with HA/CS-RhB NPs at 37 °C.

### 4.8. Competitive Binding Experiment

A competitive binding experiment was performed by treating hMSCs with mouse antiCD44 primary antibody (1.5 µg/mL). The treatment was carried out for 30 min at 4 °C followed by a recovery at 37 °C, 5% CO_2_. After 24 h, the cells were exposed to HA/CS-RhB NPs for 90 min at 37 °C. After double washing with sterile PBS, hMSCs were fixed by 4% *w*/*v* paraformaldehyde aqueous solution and 10 min incubation. Hoechst (0.5 µg/mL) were used to stain cell nuclei with, while CD44 receptors was visualized by ICC assay.

### 4.9. Transport Inhibition of HA/CS NPs Uptake

The uptake of HA/CS-RhB NPs was systematically studied by using several pharmacological inhibitors, enabling us to control different aspect of endocytosis. The inhibitors used are: Genestein, an inhibitor of tyrosine kinases liable of caveolae-mediated endocytosis; Chlorpromazine, which blocks the clathrin disassembly during clathrin-mediated endocytosis; Nocodazole, capable of disrupting microtubules; and Cytochalasin D, an actin-distrupting agent [8]. Briefly, 20,000 hMSCs were seeded onto glass slides and incubated for 24 h before the inhibition and NP treatment. After 24 h, cells were pre-treated for 30 min with the different inhibitors at the following concentrations: genistein 200 µM, nocodazole 20 µM, chlorpromazine hydrochloride 10 µg/mL and cytochalasin D 5 µg/mL. After the pre-incubation, 200 µL of HA/CS-RhB NPs suspension (100 µg/mL) was added and incubated for 90 min. At scheduled time points, all culture mediums were discarded, and hMSCs were washed with sterile PBS buffer to optimally remove non-internalized HA-RhB NPs. Finally, treated hMSCs were fixed with 4% (*w*/*v*) paraformaldehyde aqueous solution. Following the immunocytochemistry labelling and nuclei staining by 0.5 µg/mL Hoechst33258, all cells were analyzed by confocal microscopy (Leica TCSSP8, AOBS, Germany, obj mag 40×).

### 4.10. Cell Morphology Analysis

Cell morphology is one of the major aspects to be considered during the treatments because it allows us to figure out the healthy cellular status easily and quickly. Indeed, cellular morphological alterations are termed as modifications affecting the physiology of a specific cytotype, therefore suggesting pathological conditions [52].

In this study, we monitored the hMSCs’ morphology after the endocytosis inhibition treatments. Cells’ morphological evaluation was performed observing the hMSCs’ shape and behavior on the optical microscope. Moreover, the investigation was thorough in checking the cellular actin’s filaments organization. In these regards, actin’s filaments were stained taking advantage from phalloidin, a heptapeptide toxin from the mushroom (Amantia phalloides) that binds preferentially to F-actin. Briefly, treated cells and the respective controls were fixed with paraformaldehyde 4% (*w*/*v*) aqueous solution. A phalloidin-FITC solution was dropped on the cells and incubated at room temperature for 20 min and shaded from the light. Afterwards, cells were washed twice with non-sterile PBS, and nuclei were stained with 0.5 g/mL Hoechst 33258. Finally, cells’ morphology was observed by confocal microscopy (Leica TCSSP8, AOBS, Germany, obj mag 40×).

### 4.11. Statistical Analysis

Results are reported as mean ± SD calculated by at least three independent batches by ANOVA tests; Tukey’s multiple comparison test was used to reveal statistical significances. Statistical evaluations were carried out by GraphPad Prism version 6 (GraphPad Software Inc., La Jolla, CA, USA).

## 5. Conclusions

Based on the evidence of this work, two main conclusions can be drawn. Firstly, regardless of the HA used, the internalization of HA-based NPs mainly occurs by active transport exploiting the interaction between CD44. When active processes were inhibited, the NP uptake was dramatically reduced, indicating that energy use is required to internalize NPs.

Secondly, we have demonstrated the crucial role played by HA and the ability of HA-based NPs to enter cells following different endocytic pathways simultaneously. HA with the lowest Mw (280 kDa) triggers mainly a clathrin-mediated endocytosis to internalize HA/CS NPs, and moreover, it showed a slight ability to promote hMSC proliferation. HA with medium Mw (540 kDa) is liable of an unspecific NPs cellular uptake. Finally, by using HA of 820 kDa, NPs are effectively internalized via caveolae-mediated endocytosis with a more important cytoskeleton involvement.

All these outcomes are significant to accelerate the translation and use of HA/CS NPs in drug delivery, highlighting the evident influence of HA Mw on modulating the biological responses through complex effects involving different cellular pathways and influencing NPs cellular fate.

## Figures and Tables

**Figure 1 pharmaceuticals-15-00103-f001:**
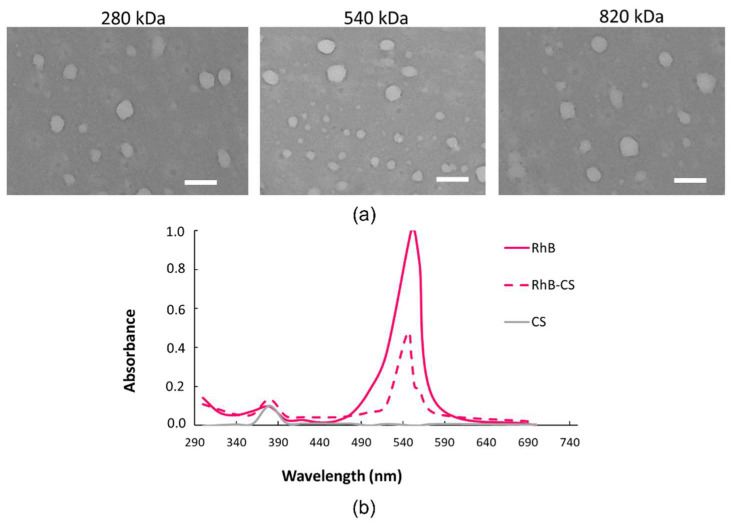
(**a**) TEM images of HA/CS NPs made of 280 kDa, 540 kDa or 820 kDa HA (scale bar = 400 nm); (**b**) UV-vis spectra scanned from 200 nm to 700 nm of CS-RhB conjugate (red dashed line), RhB (red full line) and CS (grey line).

**Figure 2 pharmaceuticals-15-00103-f002:**
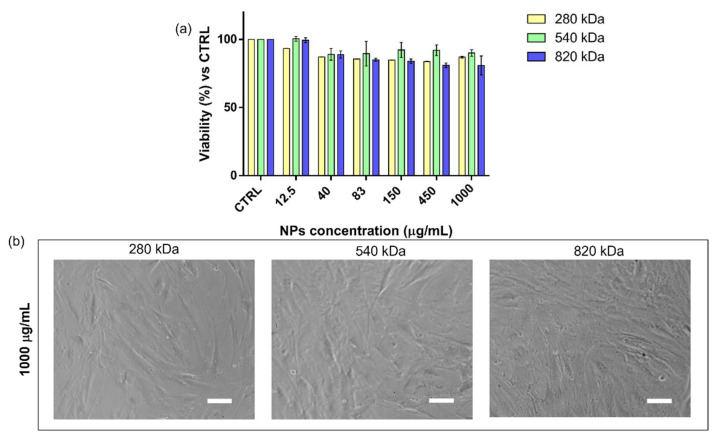
(**a**) Cytotoxicity results obtained by the incubation (24 h) of hMSCs with increasing concentration of HA/CS NPs (12.5–1000 μg/mL). Different HA Mws were used, 280 kDa (yellow bar), 540 kDa (green bar) and 820 kDa (blue bar). The graph represents the cell viability percentage compared with that of untreated cells (CTRL). (**b**) Optical microscope images of hMSCs treated for 24 h with 1000 μg/mL of HA/CS NPs (scale bar = 50 μm).

**Figure 3 pharmaceuticals-15-00103-f003:**
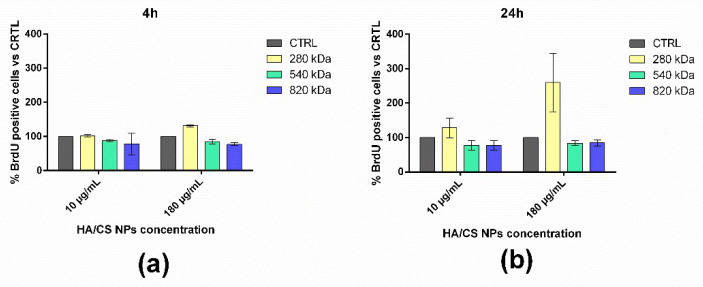
Effect on hMSCs proliferation of HA/CS NPs made of HA with Mw of 280 kDa (yellow bar), 540 kDa (green bar) and 820 kDa (blue bar) after an incubation time of (**a**) 4 h and (**b**) 24 h. CTRL (grey bar) represents untreated cells. Results are expressed as percentage of proliferating cells (BrdU positive nuclei) compared to the total viable cells.

**Figure 4 pharmaceuticals-15-00103-f004:**
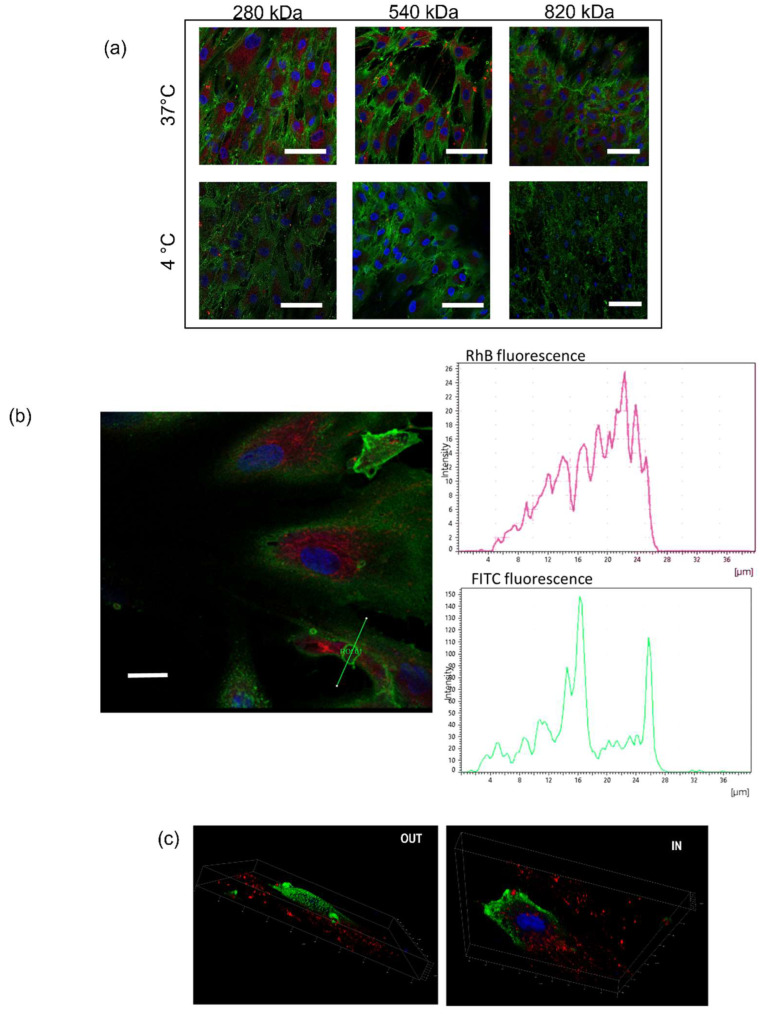
(**a**) Confocal microscopy images of hMSCs following the incubation with HA/CS-RhB NPs (20 μg for each HA) at 37 °C and 4 °C for 90 min. Red fluorescence denotes HA/CS NPs; blue Hoechst33258 dye stained DNA in the nuclei; positive expression for CD44 was highlighted by anti-CD44 primary antibody and FITC labeled secondary antibody (green fluorescence); (scale bar = 50 μm). (**b**) Confocal image of hMSCs incubated for 90 min with HA/CS-RhB NPs (280 kDa HA) and analysis of the red (RhB) and green (FITC) fluorescence intensities measured along the yellow line crossing the endocytic vesicles (scale bar = 20 µm). (**c**) Three-dimensional images of hMSCs treated with HA/CS-RhB NPs (820 kDa HA).

**Figure 5 pharmaceuticals-15-00103-f005:**
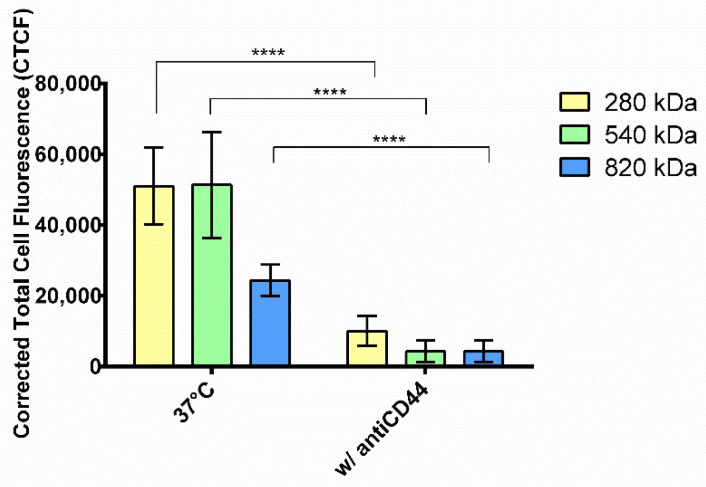
Energy dependence experiments: red fluorescent intensities of internalized HA/CS-RhB NPs after 90 min of incubation at 37 °C and 4 °C. HAs with different Mws were used: 280 kDa (yellow bar), 540 kDa (green bar) and 820 kDa (blue bar). Results are shown as mean ± SD. **** *p* value < 0.0001.

**Figure 6 pharmaceuticals-15-00103-f006:**
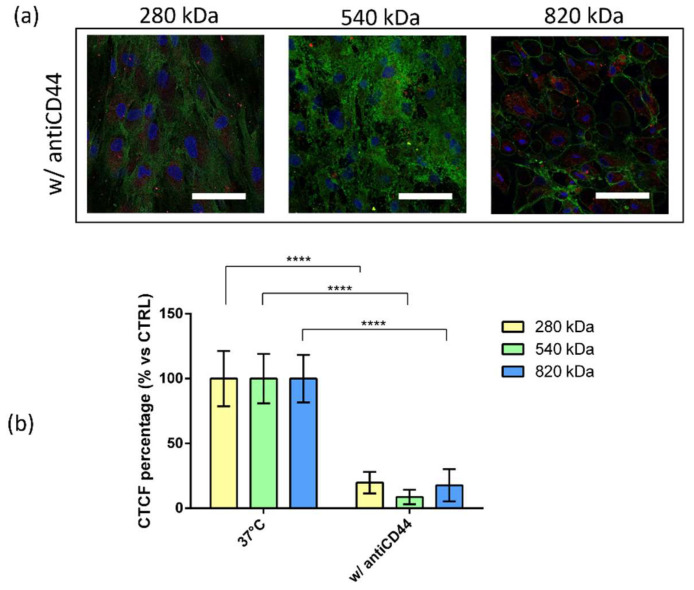
Competitive binding experiments: (**a**) Confocal microscopy images of hMSCs pretreated with monoclonal antiCD44 primary antibody and incubated with HA/CS-RhB NPs for 90 min (scale bar = 50 μm); (**b**) red fluorescent intensities of HA/CS-RhB NPs localized in the cell cytoplasm after 90 min of incubation at 37 °C with (w/antiCD44) or without (37 °C, CTRL) antiCD44 primary antibody. HAs with different Mw were used: 280 kDa (yellow bar), 540 kDa (green bar) and 820 kDa (blue bar). Results are presented as CTCF percentage mean ± SD. **** *p* value < 0.0001.

**Figure 7 pharmaceuticals-15-00103-f007:**
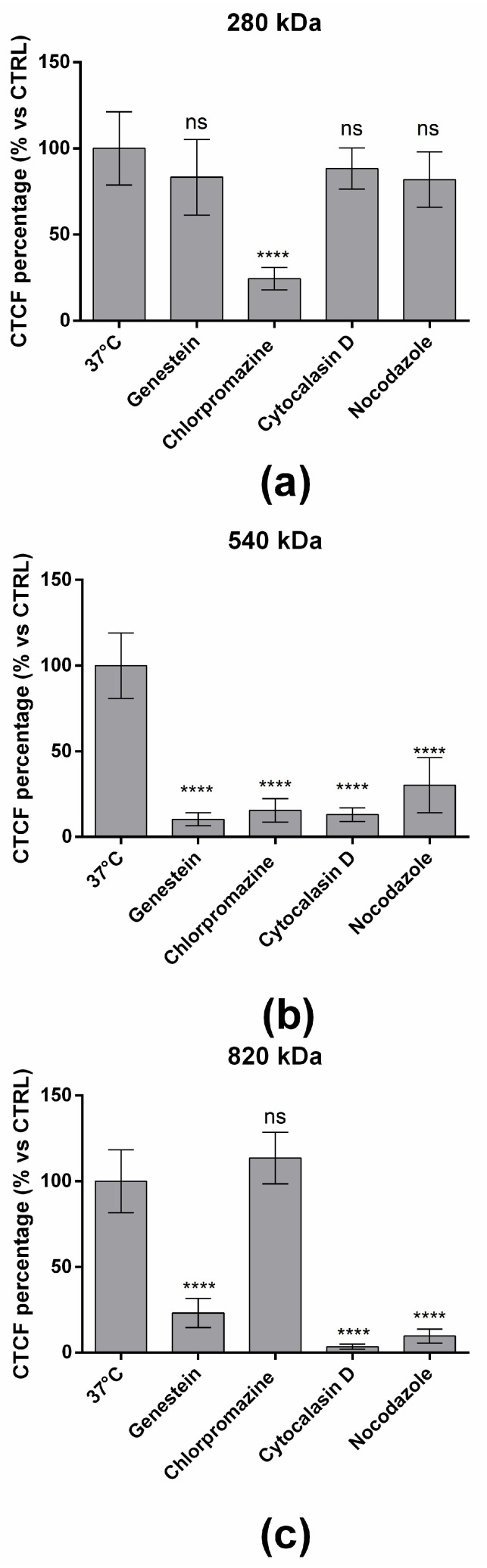
Effect of transport inhibitors on the internalization of HA/CS-RhB NPs made of (**a**) 280 kDa, (**b**) 540 kDa and (**c**) 820 kDa HA. Results are expressed as CTCF percentage mean ± SD. ns *p* value > 0.05; **** *p* value < 0.0001.

**Figure 8 pharmaceuticals-15-00103-f008:**
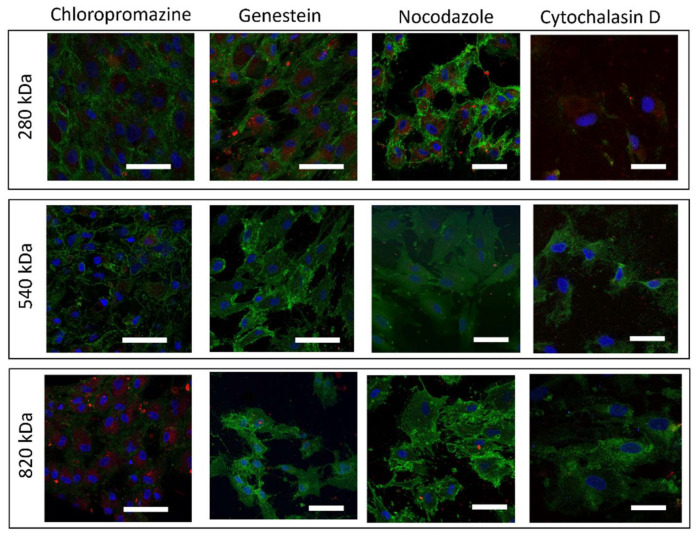
Confocal microscopy images of hMSCs after exposure to each one of the inhibitors (Chlorpromazine, Genistein, Nocodazole, Cytochalasin D) for 30 min and then exposed to HA/CS NPs (280, 540 and 820 kDa HA) for 90 min (scale bar = 50 μm).

**Figure 9 pharmaceuticals-15-00103-f009:**
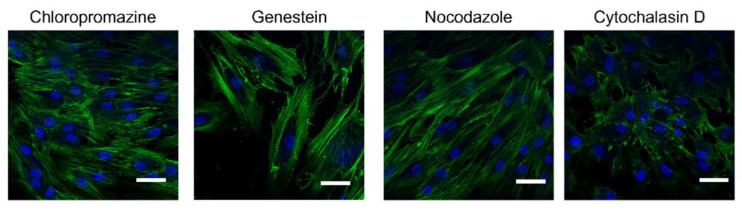
Confocal microscopy images of hMSCs showing the F-actin filaments morphology after treatment with each one of the inhibitors (Chlorpromazine, Genistein, Nocodazole, Cytochalasin D) for 30 min. Nuclei were stained by Hoechst33258 and appear blue, F-Actin was stained with FITC-phalloidin and appears green (scale bar = 20 μm).

**Table 1 pharmaceuticals-15-00103-t001:** HA and CS solutions concentrations for the synthesis of NPs based on the HA Mw used.

HA Mw (kDa)	[CS] mg/mL	[HA] mg/mL
280	0.125	0.375
540	0.067	0.200
820	0.043	0.130

## Data Availability

Data is contained within the article.

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
