# Peer review of "CD44-Targeted Carriers: The Role of Molecular Weight of Hyaluronic Acid in the Uptake of Hyaluronic Acid-Based Nanoparticles"

_pharmaceuticals, 2022, doi:10.3390/ph15010103_

Round 1

Reviewer 1 Report

The manuscript entitled “CD44-targeted carriers: the role of Hyaluronic Acid Molecular Weight in the uptake of Hyaluronic acid-based Nanoparticles” reported the effect of HA molecular weight on the efficacy and mechanism of cellular uptake of HA based nanoparticles. Although this study was only done in cellular level, three molecule weights (Mw 280, 540, 820 kDa) and several different endocytic pathways were studied. Therefore, I suggest to accept the manuscript for publication. Also it will be important to supplement more detailed discussion or prospect of future in vivo study.

Author Response

Reviewer #1:

The manuscript entitled “CD44-targeted carriers: the role of Hyaluronic Acid Molecular Weight in the uptake of Hyaluronic acid-based Nanoparticles” reported the effect of HA molecular weight on the efficacy and mechanism of cellular uptake of HA based nanoparticles. Although this study was only done in cellular level, three molecule weights (Mw 280, 540, 820 kDa) and several different endocytic pathways were studied. Therefore, I suggest to accept the manuscript for publication. Also it will be important to supplement more detailed discussion or prospect of future in vivo study.

Thank you for your positive feedback and valuable recommendations.

We are currently planned the future in vivo studies; the idea is to test HA/CS NPs in pancreatic adenocarcinoma (PDAC) models since CD44 expression level and isoform are well-recognized to contribute to pancreatic cancer cell plasticity, invasiveness, and response to therapy (Zhao et al. 2016; Ringel et al. 2001)

To accomplish this part of the project, an orthotopic PDAC mouse model will be generated by ultrasound-guided injection of PDAC cells, by using Vevo-LAZR-X system.

In this model the antineoplastic effects of nanoparticles will be evaluated by monitoring the tumor growth with 3D ultrasound B-mode imaging. Moreover, with PA imaging, the oxygen saturation (sO2) of the tumor masses will be monitored in live animals to evaluate the effects of treatments on the tumor hypoxia.

Future in vivo studies will be better pointed out in the revised manuscript and discussion section has been implemented.

References

Ringel, J, R Jesnowski, C Schmidt, HJ Kohler, J Rychly, SK Batra, and M Lohr. 2001. "CD44 in normal human pancreas and pancreatic carcinoma cell lines." Teratogenesis Carcinogenesis and Mutagenesis 21 (1): 97-106. https://doi.org/10.1002/1520-6866(2001)21:1<97::AID-TCM9>3.3.CO;2-F.

Zhao, S, C Chen, K Chang, A Karnad, J Jagirdar, AP Kumar, and JW Freeman. 2016. "CD44 Expression Level and Isoform Contributes to Pancreatic Cancer Cell Plasticity, Invasiveness, and Response to Therapy." Clinical Cancer Research 22 (22): 5592-5604. https://doi.org/10.1158/1078-0432.CCR-15-3115.

Reviewer 2 Report

  • The manuscript involves a very promising scientific point which us the use of a biocompatible and biodegradable targeting agent, HA, for CD44 receptors that are upregulated in different pathological disorders (e.g. cancer initiation).
  • The manuscript is well written and presented.
  • It is more beneficial to mention the differences between both types of targeting (passive and active) and refer how the presented work helped in achievement both targeting mechanisms.
  • In cytotoxicity study:

 From the statistical point of view, the differences between different M.wt HA based NPs should be determined  based on IC50 (the estimated NP concentration killing 50% of the cells).  IC50 should be calculated for each NP type.

Author Response

Reviewer #2:

The manuscript involves a very promising scientific point which us the use of a biocompatible and biodegradable targeting agent, HA, for CD44 receptors that are upregulated in different pathological disorders (e.g. cancer initiation).

The manuscript is well written and presented.

We would like to thank the reviewer for providing helpful comments and appreciating our work.

-  It is more beneficial to mention the differences between both types of targeting (passive and active) and refer how the presented work helped in achievement both targeting mechanisms.

According to the referee’s suggestion, the revised manuscript has been implemented.

- In cytotoxicity study: from the statistical point of view, the differences between different Mw HA based NPs should be determined based on IC50 (the estimated NP concentration killing 50% of the cells).  IC50 should be calculated for each NP type.

IC50 was higher than 1000 µg/mL for all the HAs tested since the cell viability percentage was always greater than 80%. As long as the inhibition curve did not reach the 50% of inhibition, the IC50 values cannot be empirically evaluated. However, we can perform a mathematical prediction by using sigmoidal dose-response curve fit of the experimental data. Resulted IC50 values are 3738 µg/mL, 6486 µg/mL and 2683 µg/mL for NPs with HA of 280 kDa, 540 kDa and 820 kDa respectively. To assess a suitable and robust statistical difference between the IC50 values the NPs concentration range should be broadened reaching at least a cell viability reduction of 50%.

The authors would point out that this NPs concentration range was set considering the following loading of an anticancer drug with main purpose of showing the differences between drug loaded NPs, placebo NPs and free drug also calculating the IC50 of free drug and drug loaded NPs.

This concept has been addressed in the revised version of the Manuscript.

Reviewer 3 Report

  1. Title should be modified. E.g. Molecular weight of hyaluronic acid instead of hyaluronic acid molecular weight.
  2. Manuscript should be checked for grammatical and typographical errors.
  3. Inconsistency in the use of abbreviation, make it either HA/CS NPs or HA-CS NPs.

Author Response

Reviewer #3:

- Title should be modified. E.g. Molecular weight of hyaluronic acid instead of hyaluronic acid molecular weight.

Title has been modified accordingly to reviewer’s suggestion.

- Manuscript should be checked for grammatical and typographical errors.

Manuscript has been checked for grammatical and typographical errors.

- Inconsistency in the use of abbreviation, make it either HA/CS NPs or HA-CS NPs.

Abbreviation has been checked in the revised manuscript.

Reviewer 4 Report

The article “CD44-targeted carriers: the role of Hyaluronic Acid Molecular Weight in the uptake of Hyaluronic acid-based Nanoparticles” by Chiesa et al. describes the preparation and cellular uptake mechanism of HA/CS nanoparticles with a different molecular weight of HA (Mw 280, 540, 820 kDa). This article is within the scope of this journal. However, there are several concerns that need to be fixed to justify its publication in Pharmaceutics.

My primary concerns are:

  • Many articles describe the relationship between hyaluronic acid molecular weight and active targeting efficiency (PMID: 32104479). Thus author should clearly explain the novelty of this study in the introduction section.
  • HA/CS-RhB NPs showed a mean diameter of 249 ± 13.4 nm, 183.4 ± 23.3 nm, and 235 ± 26.5 using HA with MW of 280, 540, and 820 kDa, respectively. Please provide the statistical values among particle sizes made with different MW HA.
  • The result of the effect of HA MW on hMSCs proliferation is quite interesting. What is the long-term (7-14 days) effect of HA nanoparticle exposure on hMSCs proliferation?
  • Cellular uptake study: Quantitative analysis using confocal data is not well accepted. Since nanoparticles on the cell surface will have higher fluorescence than the internalized particles. Also, it takes images of a small area of cells. Authors should quantify the fluorescence of cell lysate using fluoresce microscopy or HPLC. What percent of nanoparticles are internalized by hMSCs?
  • Please provide the molecular weight of chitosan used for this study.

Author Response

Reviewer #4:

The article “CD44-targeted carriers: the role of Hyaluronic Acid Molecular Weight in the uptake of Hyaluronic acid-based Nanoparticles” by Chiesa et al. describes the preparation and cellular uptake mechanism of HA/CS nanoparticles with a different molecular weight of HA (Mw 280, 540, 820 kDa). This article is within the scope of this journal. However, there are several concerns that need to be fixed to justify its publication in Pharmaceutics.

My primary concerns are:

- Many articles describe the relationship between hyaluronic acid molecular weight and active targeting efficiency (PMID: 32104479). Thus author should clearly explain the novelty of this study in the introduction section.

We appreciate the reviewer’s valuable comment. In literature the relationship between HA molecular weight (Mw) and active targeting efficiency has been described, above all for lipid nanoparticles and within the limits of a small panel of Mws and cell lines, but precise nature of the mechanisms involved are not clarified so far (Mizrahy et al., 2011; Qhattal et al., 2011). As long as HA/CS NPs are concerned, traditional preparation methods (bulk techniques) seemed not to affect NPs uptake, whereas different HA presentation seems to have major effects on NPs-receptor interaction (Gennari et al., 2019). More recently, HA mediated targeting should be interpreted as a more complex phenomenon of endocytic recognition rather than a cell surface event of selective binding (Rios de la Rosa et al., 2019). In this manuscript we would like to test and emphasize the effect of HA Mw on cellular uptake mechanism of HA/CS NPs synthetized by a microfluidic technique in order to provide further insights into targeted HA based NPs used for cancer therapy.

The novelty of the study has been better explained in the Introduction of the revised manuscript.

References:

Mizrahy S, Raz SR, Hasgaard M, Liu H, Soffer-Tsur N, Cohen K, Dvash R,  Landsman-Milo D,  Bremer MGEG, Moghimi SM, Peer D. “Hyaluronan-coated nanoparticles: The influence of the molecular weight on CD44-hyaluronan interactions and on the immune response”. J. Contr. Rel. 2011, 156, 231–238. doi:10.1016/j.jconrel.2011.06.031

Qhattal HSS and Liu X. “Characterization of CD44-Mediated Cancer Cell Uptake and Intracellular Distribution of Hyaluronan-Grafted Liposomes”. Mol. Pharmaceutics, 2011, 8, 1233–1246. doi:0.1021/mp200042

Gennari A, Rios de la Rosa JM, Hohn E, Pelliccia M, Lallana E, Donno R, Tirella A,and Tirelli N. “The different ways to chitosan/hyaluronic acid nanoparticles: templated vs direct complexation. Influence of particle preparation on morphology, cell uptake and silencing efficiency”. Beilstein J. Nanotechnol. 2019, 10, 2594–2608. doi:10.3762/bjnano.10.250

Rios de la Rosa JM, Pingrajai P, Pelliccia M, Spadea A, Lallana E, Gennari A, Stratford IJ, Rocchia W, Tirella A,and Tirelli N. “Binding and Internalization in Receptor-Targeted Carriers: The Complex Role of CD44 in the Uptake of Hyaluronic Acid-Based Nanoparticles (siRNA Delivery)”.  Adv. Healthcare Mater., 2019, 1901182 (11). doi:10.1002/adhm.201901182

- HA/CS-RhB NPs showed a mean diameter of 249 ± 13.4 nm, 183.4 ± 23.3 nm, and 235 ± 26.5 using HA with MW of 280, 540, and 820 kDa, respectively. Please provide the statistical values among particle sizes made with different MW HA.

Tukey’s multiple comparisons statistical analysis was used to assess size differences among the formulations obtained from HAs with different Mw (both blank and fluorescent).

 The authors thank for the clarification since there is a typing error (the mean size of NPs with HA of 540 kDa is 183.4 ± 43.3 nm). The typing error is now corrected in the revised manuscript.

- The result of the effect of HA MW on hMSCs proliferation is quite interesting. What is the long-term (7-14 days) effect of HA nanoparticle exposure on hMSCs proliferation?

The authors agree with referee that evidences on hMSCs proliferation are more evident after a long-term treatment.

However long-term effect of a single dose of HA NPs was not evaluated at the first instance because NPs are usually eliminated by kidney and liver in 24h. The authors think that a proper evaluation of the NPs long-term effects should be performed considering multiple doses of NPs thereby dose and treatment frequency should be properly selected on the base of the application. Thanks to the referee suggestion further study will be planned. Author’s consideration on long-term effect of HA/CS NPs are now added in the revised manuscript.

- Cellular uptake study: Quantitative analysis using confocal data is not well accepted. Since nanoparticles on the cell surface will have higher fluorescence than the internalized particles. Also, it takes images of a small area of cells. Authors should quantify the fluorescence of cell lysate using fluoresce microscopy or HPLC. What percent of nanoparticles are internalized by hMSCs?

The authors agree with the reviewer the confocal images elaboration is not the best choice to assess the NPs internalization however accordingly with Jonkman et al. (Jonkman et al. 2020), if proper used, confocal microscopy is a good tool to perform quantitative measurements in cells and tissues with high spatial precision.

The authors decided to use confocal microscopy because it is timesaving, allows both qualitative and quantitative data and it requires low number of cells (˜20,000). All these advantages are very useful for a rapid preliminary screening of several operation conditions (37°C, 4°C, anti-CD44 primary antibody, different transport inhibitors). When the main differences (among different inhibitors and different cell lines) will be highlighted, the experiments will be replicated and consolidated by using a higher number of cells and performing the quantification by flow cytometer.

Finally, the authors agree that confocal microscope takes images of a small area of cells and so 10 different images for the same bottom glass slide were taken. This information has been added in the revised manuscript.

Reference:

Jonkman J, Brown CM, Wright GD, Anderson KI and North AJ. 2020. "Tutorial: guidance for quantitative confocal microscopy." Nature Protocols 15 (5): 1585-1611. https://doi.org/10.1038/s41596-020-0313-9.

- Please provide the molecular weight of chitosan used for this study.

Chitosan molecular weight is 110 kDa. This information is now added in the revised manuscript.

Round 2

Reviewer 4 Report

The authors have adequately addressed my concerns, and it could be accepted for publication in Pharmaceuticals.

Author Response

Reviewer 4:

The authors have adequately addressed my concerns, and it could be accepted for publication in Pharmaceuticals.

Thanks for the positive feedback. English language and style have been checked in the revised manuscript.